Innovative multi objective optimization based automatic fake news detection

http://orcid.org/0000-0003-2756-5434 Barut Cebrail 1 cbarut@firat.edu.tr
Yildirim Suna 2
http://orcid.org/0000-0002-3513-0329 Alatas Bilal 3
http://orcid.org/0000-0002-4096-4838 Yildirim Gungor 4
1 Department of Continuing Education Center, Firat (Euphrates) University , Elazig , Turkey
2 Data Processing Department, Secretary General of Special Provincial Administration, Data Processing Department, Secretary General of Special Provincial Administration , Elazig , Turkey
3 Software Engineering, Firat (Euphrates) University , Elazig , Turkey
4 Computer Engineering, Firat (Euphrates) University , Elazig , Turkey
Datta Anwitaman
Electronic publication date: 2025 Aug 11
Publication date: 2025
Volume: 11
Electronic Location ID: e3016
Received 2024 Dec 10; Accepted 2025 Jun 19
Copyright: © 2025 Barut et al.
Copyright year: 2025
Copyright holder: Barut et al.
License: This is an open access article distributed under the terms of the Creative Commons Attribution License, which permits unrestricted use, distribution, reproduction and adaptation in any medium and for any purpose provided that it is properly attributed. For attribution, the original author(s), title, publication source (PeerJ Computer Science) and either DOI or URL of the article must be cited.
License URL: https://creativecommons.org/licenses/by/4.0/

Keywords: Multi objective optimization, Fake news detection, Metaheuristic algorithms

Funding: The authors received no funding for this work.

==============================
With the digital revolution, access to information is expanding day by day and individuals can access information quickly through the internet and social media platforms. However, in most cases, there is no mechanism in place to evaluate the accuracy of news that spreads rapidly on social media. This increases the potential for fake news to mislead both individuals and society. In order to minimize the negative effects of fake news, it has become a critical necessity to detect them quickly and effectively. Metaheuristic methods can provide more effective solutions in fake news detection compared to traditional methods. Especially in small datasets, metaheuristics are known to produce faster and more effective solutions than artificial intelligence and machine learning based methods. In the literature, the majority of fake news detection studies have focused on the optimization of a single criterion. In this study, unlike other studies, a method that enables simultaneous optimization of two criteria (precision and recall) in fake news detection is developed. In the proposed approach, an innovative solution is presented by using the Crowding Distance Level method instead of the Crowding Distance method used in the standard Non-dominated Sorting Genetic Algorithm 2 (NSGA-2) algorithm. The proposed method is tested on four different datasets such as Covid-19, Syrian war daily news and FakeNewsNet (Gossipcop). The results show that the proposed method achieves high success especially on small datasets.

Introduction

The rapid expansion of the digital ecosystem in recent years has generated an unprecedented demand for information. While this facilitates easy access to knowledge, it also enables the swift dissemination of false and misleading information, which can have detrimental effects on individuals, societies, and democratic processes. One of the most concerning aspects of fake news is its capacity to erode trust in reliable information sources. Carefully crafted and strategically timed fake news can blur the lines between fact and fiction, affecting even scientific organizations and government agencies. This phenomenon poses significant risks in critical domains such as public health, the economy, and social stability. For instance, during the COVID-19 pandemic, the spread of unsubstantiated claims regarding transmission routes and treatment methods led to serious public health consequences (Apuke & Omar, 2020). Similarly, misinformation about financial markets and investments has been linked to market manipulation, economic losses, and reputational damage (Zhi et al., 2021). Moreover, fake news often exploits existing prejudices and anxieties, exacerbating social polarization and potentially inciting social unrest (Verma, 2018). Particularly tragic are the instances of misinformation during natural disasters; for example, false information circulated during the Maraş and Antep earthquakes in Turkey deeply impacted social cohesion and underscored the critical importance of access to accurate information during crises (Canetta, 2023). These examples collectively illustrate the profound and destructive effects of fake news at both individual and societal levels.

Global events such as the COVID-19 pandemic and the Ukraine War have significantly accelerated individual’s engagement with the digital world, creating a communication environment where information can reach wider audiences in much shorter periods of time (Gottfried & Shearer, 2016). Within this expansive digital ecosystem, social media platforms—used by billions and often difficult to regulate—have become prominent sources of information alongside official and scientific websites. Current data indicate that over 60% of social media users obtain their news from these platforms. In recent years, the advancement of deep learning–based deepfake technologies, coupled with insufficient content moderation by social media companies, has heightened the risk of information pollution (Jiwtode et al., 2022). Additionally, sophisticated bots and cyborg accounts play a critical role in disseminating fake content generated by such technologies and manipulating public opinion through auto-generated comments (FactCheck, 2025; Gorwa & Guilbeault, 2020; Shahid et al., 2022). In response, major technology companies such as Google and X (formerly Twitter), as well as governmental and voluntary organizations, have initiated comprehensive efforts to combat fake news (Snopes, 2025; Teyit, 2025; Tajrian et al., 2023). However, the diversity of fake news sources and generation methods, alongside their continuous evolution, necessitates the development of more innovative and multifaceted strategies. Effectively combining different approaches is critical to addressing the complex challenge of fake content dissemination.

Fake news detection is a complex task due to the increasing sophistication of fake content, which often closely resembles genuine user-generated news. This similarity complicates detection efforts, making it essential to understand the key characteristics of fake news and the actors involved in its dissemination. Successful detection typically relies on a combination of various analytical techniques (Shahid et al., 2022; Snopes, 2025; Teyit, 2025; Tajrian et al., 2023). Among these, the following approaches are particularly notable: 1. Source Profile Analysis: this involves examining the attributes of the news source, such as discrepancies between the number of accounts they follow vs. those following them, and metadata like the account creation date. Such features can indicate suspicious or inauthentic sources.

2. Time and Propagation Analysis: by analyzing reposting rates and the timing intervals of news dissemination, this method identifies unnatural or artificially amplified propagation patterns common in fake news spread.

3. Linguistic Analysis: fake news often exhibits distinctive linguistic traits, including specific spelling errors, exaggerated fonts, or stereotypical writing styles, which can be detected through textual analysis.

4. Sentiment Analysis: fake news tends to employ exaggerated and polarizing language aimed at influencing readers’ emotions and opinions.

5. Confirmation Analysis: this technique verifies the accuracy of news content by cross-referencing it with official sources or credible news outlets.

Related works

In recent years, fake news detection has become a critical research area due to the rapid proliferation of digital content and the widespread use of social media platforms. This growing challenge has led to the development of advanced techniques for identifying and mitigating the spread of misinformation. Researchers have explored a diverse range of approaches, including traditional machine learning models, deep learning architectures, and optimization-based algorithms. These methods commonly utilize linguistic features, user behavior, and network structures to enhance detection performance. Figure 1 illustrates the classification methodologies and their interrelationships (Yildirim, 2023).

Figure 1 Classification of fake news methods.

Expert fact-checkers, who specialize in the verification of news content, employ various techniques to assess the accuracy of news stories. While these methods offer high accuracy and are relatively easy to manage, they are limited by scalability and cost, especially given the increasing volume of news items (Zhou & Zafarani, 2021). As an alternative, crowdsourcing methods harness the collective knowledge and experience of a large user base. In such systems, users can flag unreliable news sources or content whose credibility is questionable. The primary advantage of crowdsourcing lies in its pluralistic approach, incorporating diverse perspectives. For instance, Tschiatschek et al. (2018) proposed a system that analyzes user activity to detect fake news, combining expert verification with crowdsourced input. This hybrid approach enhances accuracy and scalability in large-scale news feeds by preventing the dissemination of content labeled as fake.

Machine learning techniques have shown significant promise in fake news detection (Baarir & Djeffal, 2020). These models can process vast amounts of data efficiently, enabling real-time analysis. For example, Stitini, Kaloun & Bencharef (2022) applied the Naive Bayes algorithm to Facebook news, achieving 74% accuracy. While these results are encouraging, they also indicate room for improvement. Recommendation systems represent another line of research, aiming to evaluate the credibility of content and guide users toward reliable news. Although such systems may enhance accuracy through user feedback, they risk creating filter bubbles and pose privacy concerns. Stitini, Kaloun & Bencharef (2022) developed a model that informs users about the level and type of misinformation, increasing awareness and reducing its impact. Natural language processing (NLP) is crucial in fake news detection, enabling the extraction of meaning from text through linguistic analysis. Oshikawa, Qian & Wang (2020) and Devarajan et al. (2023) demonstrated the effectiveness of NLP-assisted artificial intelligence methods, with (Devarajan et al., 2023) reporting an average accuracy of 99.72% and an F1-score of 98.33% across three datasets. Graph-based approaches, on the other hand, model relationships between users, content, and sharing patterns to analyze the veracity of news on social networks. Zhou & Zafarani (2021) and Shu, Wang & Liu (2019) proposed methods that simultaneously model publisher-news and user-news interactions, showing improved detection performance by integrating content and social dynamics.

Deep learning techniques, particularly those employing neural networks, have been widely adopted in the literature. Sastrawan, Bayupati & Arsa (2022) combined Convolutional Neural Network (CNN), Bidirectional Long Short-Term Memory (Bi-LSTM), and Residual Network (ResNet) architectures with pre-trained word embeddings for fake news detection across four datasets, finding that Bi-LSTM outperformed other architectures. Güler & Gündüz (2023) developed a deep learning-based system integrating Word2Vec with CNN and Recurrent Neural Network (RNN)-LSTM, achieving high accuracy in both English and Turkish datasets. These results underscore the effectiveness and language-independence of deep learning approaches in fake news detection. Hybrid methods have also gained traction, combining the strengths of multiple techniques. Ruchansky, Seo & Liu (2017) proposed a model that not only outperformed existing methods in accuracy but also extracted meaningful user data. More recently, transformer-based models have significantly advanced fake news detection by capturing complex textual and contextual patterns. These models analyze relationships not only at the word level but also across sentences and paragraphs. When combined with social media data, transformers enable comprehensive detection by examining propagation behaviors and user interactions. Surveys such as Ruffo et al. (2023) highlight the growing effectiveness of these models. Palani, Elango & Viswanathan (2022) integrated Bidirectional Encoder Representations from Transformers (BERT) for text analysis and Capsule Network (CapsNet) for visual data, achieving 93% and 92% accuracy on the Politifact and Gossipcop datasets, respectively—surpassing the SpotFake+ model. Likewise, Pavlov & Mirceva (2022) fine-tuned BERT (Devlin et al., 2019) and Robustly Optimized BERT Pretraining Approach (RoBERTa) (Liu et al., 2019) models on COVID-19-related tweets, achieving superior results in accuracy, recall, and F1-score.

Optimization-based techniques are also prevalent in fake news detection. Ozbay & Alatas (2021) adapted Grey Wolf Optimization (GWO) and Salp Swarm Optimization (SSO) algorithms for the fake news detection problem, demonstrating efficiency and improved accuracy compared to other artificial intelligence methods. Zivkovic et al. (2021) developed an optimization-based method for detecting Covid-19-related fake news, reporting superior results over alternative techniques. These findings highlight the potential of optimization algorithms to enhance accuracy in fake news detection. In NLP, methods such as word embeddings, transformers, sentence transformers, and document embeddings are used to represent text in vector form, allowing models to better capture contextual meaning. While transformer-based models perform well on large datasets, they may risk overfitting on smaller datasets and require substantial computational resources. To address these challenges, various optimization techniques have been proposed. Notably, multi-objective optimization methods have been employed to balance model performance across different metrics. Ensemble methods, which combine information propagation and social network data, play a significant role in capturing the relational and dynamic aspects of fake news dissemination. While many studies focus on propagation patterns or network structures, some have explored balancing different performance metrics using multi-objective optimization. These approaches are particularly advantageous when comprehensive user interaction data or network topology is unavailable, as they can operate effectively on text-based datasets alone.

In this study, we propose a novel fake news detection framework that uses the Non-dominated Sorting Genetic Algorithm 2 (NSGA-2) to simultaneously optimize precision and recall. Unlike conventional transformer-based approaches, which often focus on a single metric, our method produces Pareto-optimal solutions that offer decision-makers a spectrum of trade-offs and improved flexibility. To enhance diversity and robustness, we introduce an improved Crowding Distance level (CDL) strategy that maintains population variety throughout the optimization process. While most optimization-based studies in this field address fake news detection as a single-objective problem, our formulation embraces the inherent conflict between precision and recall, providing a richer and more adaptable solution space. The resulting interpretable machine learning framework generates multiple Pareto-optimal models based on user-defined criteria, offering valuable insights and more nuanced decision-making capabilities. By integrating advanced text analysis techniques with an improved NSGA-2 framework, our method achieves superior performance compared to conventional models.

The purpose and novel aspects of the study are briefly described in “Related Works”. Details of the proposed model, text mining steps, and performance evaluation metrics are presented in “Methodology”. “Datasets” includes general information about the datasets as well as details like Exploratory Data Analysis (EDA) and similar preliminary analyses. “Experiments” presents the experimental results obtained by several algorithms implemented on the datasets. “Results” discusses the results derived from the current study, highlighting their implications and limitations.

Purpose and innovative aspects of the study

The use of metaheuristic algorithms in fake news detection provides several distinct advantages. While deep learning-based approaches have demonstrated strong performance on large and complex datasets, their effectiveness tends to decrease when applied to small or imbalanced datasets (Keshari et al., 2020). In contrast, metaheuristics are capable of producing efficient and logical solutions even under such constrained conditions (Barut, Yildirim & Tatar, 2024; Risvanli et al., 2025). Moreover, they often deliver faster results compared to deep learning-based methods (Kudela, 2022). Traditional machine learning techniques typically rely on manual intervention for feature selection and model tuning, and they may become trapped in local optima, limiting performance. Metaheuristics, by virtue of their global search capabilities, offer a more comprehensive exploration of the solution space, which helps to overcome such limitations. Additionally, their adaptability makes them particularly suitable for dynamic problems like fake news detection, where underlying data patterns frequently change. Unlike static models, metaheuristics can adjust to new data without requiring extensive retraining.

In this study, we propose the Crowding Distance Level (CDL) method—an enhancement of the widely utilized Non-dominated Sorting Genetic Algorithm 2 (NSGA-2). While the standard NSGA-2 employs the Crowding Distance (CD) technique to maintain diversity among solutions, it often falls short in preserving a well-distributed Pareto front. The CDL method introduces a systematic ranking strategy that ensures more balanced selection and improved solution diversity. This modification is particularly effective in small and imbalanced datasets. Furthermore, we adopt a multi-objective optimization approach that simultaneously maximizes precision and recall—a strategy that, to the best of our knowledge, has not been previously applied in the context of fake news detection.

Key contributions of the study

This study is the first to apply multi-objective optimization in fake news detection, addressing the trade-off between precision and recall.

The NSGA-2 algorithm is implemented in this domain for the first time.

An improved version of the CD is proposed, offering more diverse and balanced Pareto-optimal solutions than the standard CD method.

The proposed approach achieves higher accuracy compared to conventional methods under constrained data conditions.

The method produces a range of Pareto-optimal solutions, allowing stakeholders to select outcomes based on their specific priorities.

Methodology

In this section, the fundamental operation of the proposed multi-objective metaheuristic optimization method is explained in detail. To provide a clear understanding, the section begins with an overview of the text preprocessing steps and the principles of multi-objective optimization. Following this, the core algorithm employed in the study—the NSGA-2—is introduced. Lastly, the most innovative component of the proposed approach, the CDL method—developed as an enhancement to the conventional CD mechanism used in NSGA-2—is described, highlighting its working principles and advantages.

Text preprocessing

Text preprocessing methods include various steps such as tokenization, stop-word cleaning, stemming, and feature vectorization and are widely used in different architectures (Yildirim, 2023). Although these techniques are not new, they play a critical role in optimizing the feature space for various detection approaches. The textual content on the internet varies significantly from short texts of a few words to long articles. This variability requires appropriate preprocessing to ensure that it is handled effectively by the method used. In this study, the dataset goes through a cleaning process where numbers, irrelevant characters, and redundant words are filtered out. To minimize the size of the feature space, redundant words are removed and stem words are identified based on their frequency in the dataset. For word weighting, the double normalization K-term frequency method with K = 0 is applied, which normalizes term frequencies by dividing each term’s frequency by the maximum term frequency in the document (Jones, 1972). This technique ensures that the term frequency values are properly normalized, preventing the high-frequency terms from dominating while preserving their relative importance. The rationale for using term frequency (TF) word weighting over alternative methods is that it provides a simple and direct measure of the importance of words in a document, allows model results to be easily interpreted, is computationally efficient, especially on large datasets, and provides a basic representation of text data. We acknowledge the advantages of methods such as word embedding, transformer embedding, sentence transformer, document embedding, and expert blending, which have been proposed by experts in the field. These methods have the potential to capture semantic relationships between words and sentences more effectively, and can perform better on more complex tasks. However, implementing these methods requires more computational resources and increases the complexity of the model. In the scope of this article, we prefer TF due to its simplicity and efficiency. The entire process is performed as shown in Eq. (1). Here Wi weight of the Word i, Fi the number of times the word ioccurs in the dataset, Fmax is the maximum number of repetitions. It is important to eliminate words with a small number of repetitions in the dataset if they are below a certain threshold value in order to reduce the number of attributes to be used in the search space. Finally, the words in the search space are scanned in each record in the dataset. If the word is in an appropriate record, it is evaluated as 1 or 0. Thus, the dataset is transformed into a search space of 1’s and 0’s.

(1) Wi=K+(1−K)×FiFmax.

Next, the population is created and the fitness value of each individual is calculated. When calculating the fitness values of the candidates, each candidate is evaluated for each record in the edited dataset. Two criteria are taken into account at this stage. The first criterion is whether the similarity (Sim→X) ratio between the candidate values and the corresponding record is greater than a predefined threshold value (τ). Various methods can be used for this. In this study, the similarity between two binary vectors is realized using the and operator. The second criterion is whether the class of the related record is the same as the candidate class. These two criteria are considered together as shown in Table 1. Thus, the candidate can calculate the current true positive (TP), true negative (TN), false positive (FP) and false negative (FN) values. After repeating this process for all records in the dataset, Eq. (2) to calculate the candidate’s fitness values. The values of the best candidate found at the end of the iterations can provide the precision and recall metrics.

Table 1 Checking whether the registration is the same as the candidate class.

Condition 1	Condition 2	Process	
If Sim→X≥τ	the class of the data searched = = the class of the data class of data in dataset	Increase TP by 1	
If Sim→X≥τ	the class of the data searched ! = the class of the data class of data in dataset	Increase FP by 1	
If Sim→X<τ	the class of the data searched = = the class of the data class of data in dataset	Increase FN by 1	
If Sim→X<τ	the class of the data searched ! = the class of the data class of data in dataset	Increase TN by 1	

(2) Prec=TPP+FPRec=TPTP+FN.

In this study, precision and recall were chosen as evaluation metrics because they provide a more comprehensive assessment of model performance on imbalanced datasets than relying solely on a single metric such as accuracy or the F1-score. Using both precision and recall allows for a nuanced understanding of the model’s ability to balance false positive and false negative rates. While the F1-score combines precision and recall into a harmonic mean and accuracy measures overall correctness, evaluating precision and recall separately offers deeper insights into specific error types—an essential consideration when dealing with imbalanced data.

Multi/Many objective optimization (MOO)

Multi-objective optimization involves the simultaneous optimization of multiple conflicting objectives. Unlike traditional single-objective optimization, which seeks a single best solution by maximizing or minimizing one objective, multi-objective optimization aims to find a balance among competing objectives. For example, one might want to minimize the cost of a vehicle while maximizing its luxury features, or improve vehicle performance while reducing fuel consumption. When the number of conflicting objectives is fewer than four, the problem is classified as multi-objective; if it involves four or more objectives, it is referred to as many-objective optimization. Multi-objective optimization methods are generally categorized into three classes based on how they incorporate decision-maker preferences: 1. A priori methods require the decision maker to specify preferences or priorities among objectives before solving the problem. Techniques such as linear scalarization, lexicographic ordering, and auxiliary function methods transform the multiple objectives into a single objective by assigning weights or ranks. While this approach simplifies the problem, it risks excluding parts of the solution space and may reflect subjective preferences that limit generality.

2. A posteriori methods do not involve decision-maker preferences during the search process. Instead, they generate a set of Pareto-optimal solutions representing trade-offs among objectives, where no objective can be improved without worsening another. The decision maker then selects the most suitable solution from this set based on their context. This approach provides a comprehensive view of the solution space without bias toward any particular objective.

3. Progressive methods iteratively incorporate decision-maker preferences during the optimization process to converge toward the most preferred solution. Preferences are elicited gradually, allowing for refinement and adaptation as more information becomes available.

The choice of method depends on the problem characteristics, the decision maker’s role, and the relative importance of objectives. As the number of objectives increases, the complexity of finding and representing optimal solutions grows, especially in many-objective problems. In particular, a posteriori methods, which do not require prior preference information, are advantageous for producing more general and diverse solution sets in complex scenarios. Metaheuristic algorithms are widely employed in multi-objective optimization due to their versatility, global search capability, population-based approach, and adaptability to complex problems. Commonly used multi-objective metaheuristics include: NSGA-2 (Deb et al., 2002), Non-Dominated Sorting Genetic Algorithm-3 (NSGA-3) (Deb & Jain, 2014) Multi Objective Particle Swarm Optimization (MOPSO) (Coello & Lechuga, 2002), Multi Objective Genetic Algorithm (MOGA) (Murata & Ishibuchi, 1995), Strength Pareto Evolutionary Algorithm 2 (SPEA2) (Zitzler, Laumanns & Thiele, 2001), Multi Objective Ant Colony Optimization (MOACO) (Angus & Woodward, 2009), Multi Objective Particle Swarm Optimization incorporating Crowding Distance (MOPSO-CD) (Kumar et al., 2022).

NSGA-2

Metaheuristic methods are general-purpose optimization techniques designed to produce near-optimal solutions for complex problems that are difficult or infeasible to solve using deterministic approaches within a reasonable time frame. These methods are not tailored to specific problem types, making them highly adaptable across a wide range of application domains (Barut, Yildirim & Tatar, 2024). Their primary objective is to improve solution quality by efficiently exploring the solution space while avoiding entrapment in local optima. Among these methods, the NSGA-2 stands out as an advanced multi-objective optimization algorithm, developed as an enhancement of the original NSGA algorithm (Srinivas & Deb, 1994). NSGA-2 introduces significant improvements, including a more efficient non-dominated sorting procedure, the use of a CD mechanism for diversity preservation, refined parameter settings, and reduced computational complexity. These enhancements make NSGA-2 particularly effective in identifying Pareto-optimal solutions in multi-objective optimization problems.

The working principles of NSGA-2 start with the random generation of an initial population. Each individual in this population is treated as a randomly generated news item and evaluated as true or false. Each individual has attributes equal to the number of attributes in the dataset, and the initial population (P0) is randomly generated from these individuals. After the initial population is determined, a dominance check is performed among all individuals in the population.

Dominance control

In the NSGA-2 algorithm, after the initial population is created, a dominance check is performed among the individuals in the population. In an N-dimensional population, one goal of each individual is compared with the corresponding goal of other individuals to identify non dominated solutions. For example, suppose we are comparing two solutions in the population. Let us call these two solutions with two objectives A = {a1. a2} and B = {b1. b2} respectively. If (a1≤b1anda2≤b2)conditions and at the same time (a1<b1ora2<b2). Then solution A dominates solution B. This dominance check is performed for all individuals in the population. As a result of the dominance check, all solutions that do not dominate each other are grouped into lists. In the NSGA-2 algorithm, the first of these dominance lists contains the best solutions. The dominance lists are used in the next stage of the algorithm by ranking the solutions according to their importance.

Crowding distance

Although there is a prioritization between the lists obtained as a result of the dominance check, there is no ranking between the solutions within the same list. NSGA-2 uses the CD method to prioritize between solutions in the same list. This method aims to preserve the diversity of individuals in the population and helps to decide which individuals to select in the next generation.

CD is defined as the Euclidean distance between a solution on the pareto front and its two neighbors, as calculated using the formula in Eq. (3). Solutions with higher CD values in the pareto front are usually selected for use in subsequent generations. Thus, the CD method increases the diversity in the population, resulting in a more balanced distribution of solutions in the Pareto front.

(3) CDi=∑k=1M|Fi+1m−Fi−1m|Fmaxk−Fmink.

Genetic operators

The production of new individuals is a process that takes place using the genetic information of individuals in an existing population. This process involves the creation of new offspring individuals by combining the traits of two or more parental individuals. In the literature, various methods have been proposed for these processes. The NSGA-2 method frequently uses Simulated Binary Crossover (SBX) and mutation methods in genetic operations.

In the NSGA-2 algorithm, two solutions are randomly selected from the population to generate new individuals. Generally, these solutions are chosen among the individuals in the highest pareto front. If individuals from the same pareto front are to be selected, solutions with a high CD value are preferred. However, in rare cases, there may be solutions from the same front with the same CD value. In this case, which individual to select can be randomized according to user preference.

The SBX method produces new individuals by combining the genetic information of two parental individuals while maintaining population diversity. In this process, two child individuals are created from two parent individuals. The genetic values of the new individuals are designed to be close to those of the parental individuals. To determine the distribution of the children, the spread factor ( β) is used and this value is defined in Eq. (4). The similarity of new individuals to their parents is determined by the constant ( φ∈R+). A high value of φ increases the similarity of the new individuals to the parent individuals. During the crossover process, the probability distribution β∈[0,∞] is defined in the range and calculated as expressed in Eq. (5). The new individuals produced at the end of this process are calculated using Eqs. (6) and (7). This process ensures that new individuals are produced in a way that is genetically balanced and maintains the diversity of the population.

(4) β=|c1−c2p1−p2|.

(5) P(β)={0.5×(φ+1)×βφifβ≤1.0.5×(φ+1)×1βφ+2otherwise

(6) sv1=0.5×(1+βh)×pr1+(1−βh)×pr2.

(7) sv2=0.5×(1−βh)×pr1+(1+βh)×pr2.

One of the methods used to generate new individuals in the NSGA-2 method is mutation. Mutation increases the probability of generating alternative solutions close to an existing solution by making small changes to it. This process is calculated as shown in Eq. (8). During the mutation process, yn(m) represents the individual in the m’th population. The change of this individual occurs between the upper and lower limits expressed yn(U) and yn(L). αn parameter is defined in Eq. (9). In this calculation process, k is a randomly chosen value in the range [0,1], γ is used as a parameter that controls the effect and diversity of the mutation. Mutation increases the diversity of individuals in the population and contributes to a wider distribution of solutions on the pareto front.

(8) yn(m+1)=yn(m)+(yn(U)−yn(L))×αn.

(9) αn={(2×k)1γ+1−1ifk<0.5.1−(2×(1−k))1γ+1otherwise

Crowding distance level

The most innovative contribution of this study is the introduction of the CDL method as an enhancement to the conventional CD mechanism employed in the NSGA-2 algorithm. The standard CD method aims to maintain solution diversity by calculating the distance between a solution and its two nearest neighbors on the Pareto front, favoring those with higher CD values for selection into the subsequent generation. However, this approach presents several limitations: it may overlook globally optimal solutions by relying solely on local distance information.

It can inadvertently emphasize one or a few objective functions, thereby reducing the diversity of the final Pareto front.

In cases where multiple solutions share identical CD values, selection is often carried out randomly, which may compromise the balance and representativeness of the chosen solutions.

To address these limitations, the proposed CDL method introduces a systematic classification and ranking process designed to ensure a more equitable selection of solutions across the Pareto front. Unlike the CD approach, CDL eliminates the dependence on random selection by incorporating a more structured and objective evaluation mechanism. As a result, CDL enhances the diversity and quality of the final solution set, offering decision-makers a broader and more representative array of pareto-optimal solutions. This improvement significantly contributes to the robustness and effectiveness of the multi-objective optimization process. The basic operation of the CDL method, which is proposed to overcome the shortcomings of the CD method, can be summarized as follows:

all solutions on the pareto front A={A1.A2.….An} set and the solutions for which the CDL value calculated is initially B={} are considered as sets. The CDL method, like the CD method, prioritizes solutions at the extremes of the pareto front. These prioritized solutions are removed from set A and added to set B and their CDL value is assigned 0. As a result of this process, B={A1An−1} happens. Then, an iterative process is started: The total Euclidean distance of each solution in set A to all solutions in set B is calculated as shown in Eq. (11). The Euclidean distance between two solutions is defined in Eq. (10). As a result of this calculation, the solution with the largest total Euclidean distance in cluster A is identified, this solution is removed from cluster A and added to cluster B with one more CDL than the previous CDL level (e.g., the solution to be selected in the next iteration will have a CDL of 1, since it has an initial CDL of 0). This process continues until there are no solutions left in set A. When all solutions are moved to set B, all of these solutions have a CDL value. In Eq.(10) i is a solution in set A, j value refers to a solution in cluster B. The proposed method prioritizes solutions with low CDL values and moves them to the next generation.

(10) F(i,j)=∑t=1n⁡(xt−yt)2.

(11) SCDi=∑j=1||B||⁡Fi,j.

Figure 2 illustrates the results of evaluating a Pareto front consisting of nine randomly generated solutions using both CD and CDL methods. Solutions that share the same CD or CDL value are represented with the same color for visual clarity. In Fig. 2A, it can be observed that six solutions are grouped into three pairs with identical CD values. This highlights a limitation of the CD method, as it fails to sufficiently distinguish between solutions with similar characteristics, often leading to random selection when CD values are equal. On the other hand, Fig. 2B demonstrates that the CDL method assigns distinct levels to all solutions—except those initially evaluated at the same dominance level—providing a more nuanced and detailed ranking. Unlike CD, CDL introduces a structured classification process that enhances the differentiation among solutions.

Figure 2 Comparison of approaches.

Proposed approach and runtime analysis

The proposed method is a combination of the steps mentioned in the previous sections and the pseudocode of the method is given in Algorithm 1. Since we use the standard NSGA-2 algorithm as the basis in our study, our time complexity is the same as NSGA-2. The theoretical time complexity of NSGA-2 is O(M×N2), where M is the number of objective functions and N is the population size. The improvement used in our study is an improved version of Crowding Distance calculation and does not change the overall time complexity. Therefore, the time performance of the method is at the same level as the standard NSGA-2.

Algorithm 1 Pseudocode of the proposed method.

   Input:	
     Dataset D: Fake news dataset (text data)	
     Population Size P: Number of candidate solutions	
     Max Iterations I: Maximum number of evolutionary iterations	
     Cross over Rate C: Probability of crossover operation	
     Mutation Rate M: Probability of mutation operation	
   Output:	
     Optimized Solutions S: Set of optimized solutions	
1. Preprocessing:	
   1.1.  Clean the text: Remove unnecessary characters and stop-words.	
   1.2.  Perform stemming and word frequency analysis.	
   1.3.  Create feature vectors and convert data into binary format.	
2. Population Initialization:	
   2.1.  Generate P random binary vectors, each representing a candidate feature selection solution.	
   2.2.  Each binary vector (individual) represents a subset of selected features.	
3. Evaluate Fitness:	
   3.1.  Compute precision and recall.	
   3.2.  Evaluate candidate solutions based on multi-objective optimization goals.	
4. While termination criterion is not met (iteration < I):	
   4.1.  Dominance Sorting: Rank solutions based on dominance.	
   4.2.  Selection:	
        4.2.1.    Select top-ranked solutions.	
        4.2.2.    Use Crowding Distance Level (CDL) to maintain solution diversity.	
   4.3. Crossover and Mutation:	
        4.3.1.   Perform crossover based on the crossover rate C.	
        4.3.2.   Apply mutation with probability M to introduce small variations.	
        4.3.3.   Compute the fitness values of new individuals.	
        4.3.4.   Select the best solutions for the next generation.	
5. Return optimized solution set S.	

Datasets

This section provides basic information about the datasets used in the study. The general characteristics, structure and attributes of the datasets are explained; and the findings obtained through Exploratory Data Analysis (EDA) are also included. Experiments were conducted using four datasets: Covid-19 (Koirala, 2021), Syrian War (Salem et al., 2019), General News (Ahmed, Traore & Saad, 2018), and FakeNewsNet (Shu et al., 2020). These datasets include both small and large-scale news samples with varying class distributions. The FakeNewsNet dataset, widely used in misinformation research, was particularly included to assess generalizability on real-world data. Summary information about the datasets used in the experiments is presented in Table 2.

Table 2 Characteristics of the datasets used.

Dataset	Number of data in the data set	True news	Fake news	Number of word	Class	
Covid 19	3.119	2.061	1.058	134	True/False	
Syria war	804	426	378	109	True/False	
General news	44.858	21.417	24.441	136	True/False	
FakeNewsNet (Gossipcop)	22.125	16.790	5.335	147	True/False	

Covid-19 dataset

The dataset used in this study covers news content related to Covid-19. Exploratory Data Analysis (EDA) was applied to understand the structural features of the dataset and to examine the word patterns used in the news texts. The analysis results obtained are presented in Fig. 3. Figure 3A shows the frequency distribution of words in all news texts, highlighting the most frequently used terms. This visualization provides a general idea about the dominant themes and word structures in the dataset. Figure 3B shows the distribution of fake and real news, allowing the observation of the class imbalance situation, which is important for the training and evaluation of the model.

Figure 3 Analysis of Covid-19 dataset.

Syrian war dataset

The dataset used in this study includes news content related to the Syrian War. Exploratory Data Analysis (EDA) was applied to understand the structural features of the dataset and to examine the word patterns used in the news texts. The analysis results obtained are presented in Fig. 4. Figure 4A shows the frequency distribution of the words in the news texts, revealing the prominent terms in the dataset. Figure 4B shows the distribution of fake and real news samples, allowing the evaluation of whether there is a class imbalance.

Figure 4 Analysis of Syrian war dataset.

General news dataset

The dataset used in this study covers daily news content. Exploratory Data Analysis (EDA) was applied to understand the structural features of the dataset and to examine the word patterns used in the news texts. The analysis results obtained are presented in Fig. 5. Figure 5A shows the frequency distribution of the words in the news texts, revealing the prominent terms in the dataset. Figure 5B shows the distribution of fake and real news samples, allowing the evaluation of whether there is a class imbalance.

Figure 5 Analysis of general news dataset.

FakeNewsNet GossipCop dataset

This study utilizes the FakeNewsNet GossCop dataset, which contains news articles labeled as fake or real. To understand the general structure of the dataset and to analyze word usage patterns within the news texts, Exploratory Data Analysis (EDA) was first performed. The results of this analysis are presented in Fig. 6. Specifically, Fig. 6A visualizes the frequency distribution of words in the news texts, highlighting the prevalence of certain terms within the dataset. Figure 6B shows the distribution of fake and real news articles, allowing for the evaluation of potential class imbalance.

Figure 6 Analysis of general news dataset.

Experiments

The performance of the proposed method was evaluated against several baseline machine learning models, including Support Vector Machines (SVM), RNN, Decision Trees (DT), Naive Bayes (NB), CNN, and Bi-LSTM, BERT and ROBERTa. All datasets were preprocessed and transformed into binary representations, as detailed in “Methodology”. A 45% threshold was applied to determine feature inclusion within the search space. As a result, the number of binary features (including the class label) was 135 for the Covid-19 dataset, 110 for the Syrian War dataset, 137 for the General News dataset, and 148 for the GossipCop subset of the FakeNewsNet dataset. Experiments were performed on all datasets using the five-fold cross-validation method. Statistical results of the cross-validation process for each dataset are presented separately and performance metrics in the training and testing stages are reported in detail. Performance comparisons were made with Support Vector Machines (SVM), Recurrent Neural Networks (RNN), Decision Trees (DT), Naive Bayes (NB), Convolutional Neural Networks (CNN) and Bi-directional Long Short-Term Memory Networks (Bi-LSTM) methods, which are widely used and currently accepted machine learning algorithms in the literature, as well as BERT and RoBERTa methods. All parameters used in the experiments are presented in detail in Table 3. These values represent the optimal hyperparameters obtained as a result of the hyperparameter analysis performed prior to the experiments. The hyperparameter settings of the methods compared in Table 4 provided. Since the proposed system works with a supervised learning approach, the datasets are divided into training and test subsets and the models created with the training data are evaluated separately on the test data. All experiments were performed on a computer with an Intel(R) Core(TM) i5-10400 CPU @ 2.90 GHz processor and 8 GB RAM.

Table 3 Experiments parameters.

Parameters	Value	
Independent experiments for each data set	10	
K fold validation value	5	
N	20	
λ	0.5	
τ	((Word Number of DataSet) * 45)/100	
γ	5	
φ	5	
Crossover probability	0.8	
Mutation probability	0.1	
Max. Iteration	100	

Table 4 Hyperparameters used for the machine learning models.

Model	Hyperparameter	Value	
Support Vector Machine (SVM)	Kernel	Radial Basis Function (RBF)	
Regularisation parameter (C)	1	
Gamma	Scale	
Tolerance	0.001	
Recurrent Neural Network (RNN)	Number of layers	2	
Hidden units per layer	128	
Activation function	ReLU	
Optimiser	Adam	
Learning rate	0.001	
Batch size	32	
Dropout rate	0.3	
Epochs	50	
Decision Tree (DT)	Criterion	Gini inequality	
Maximum depth	20	
Minimum samples split	2	
Minimum samples per leaf	1	
Naïve Bayes (NB)	Model	Multinomial Naïve Bayes	
Smoothing parameter (alpha)	1	
Convolutional Neural Network (CNN)	Number of convolutional layers	2	
Kernel size	3 × 3	
Pooling	MaxPooling (2 × 2)	
Fully connected units of layers	128	
Activation function	ReLU	
Optimiser	Adam	
Learning rate	0.001	
Batch size	32	
Dropout rate	0.4	
Epochs	50	
Bidirectional Long Short-Term Memory (Bi-LSTM)	Number of LSTM layers	2	
Cached units per layer	256	
Activation function	Tanh	
Optimiser	Adam	
Learning rate	0.001	
Batch size	32	
Dropout rate	0.3	
Epochs	50	
BERT	Logistic regression	max_iter	1,000	
solver	‘lbfgs’	
penalty	‘l2’	
C	1	
fit_intercept	True	
class_weight	None	
ROBERTa	Hyperparameter	Pretrained model	roberta-base	
num_labels	2	
Batch size	16	
Learning rate	2.00E−05	
Optimizer	AdamW	
Epochs	3	
Max sequence length	128	

Covid-19 experiments

In this section, a series of experiments were conducted using a dataset comprising news articles related to Covid-19. These preliminary analyses yielded significant insights into the dataset’s composition and served as a foundational step for the subsequent modeling processes. Furthermore, Fig. 7 displays the top 10 most influential word stems and their average weights, derived from a total of 135 stems, in the classification of fake and real news during the Covid-19 experiments. The analysis revealed that the stem “isol” had the highest frequency in the fake news class, whereas “unit” was most frequent in the real news class. Additionally, the word stem “issu” was notably prominent in the fake news class, unlike in the real news. Despite these differences, a high degree of similarity was observed in the frequently occurring terms across both classes, indicating overlapping linguistic features that may pose challenges in classification tasks.

Figure 7 The ten word stems with the highest average weights in the Covid-19 data set experiments (A) Fake (B) True.

The results of the experiments conducted using the Covid-19 dataset are presented in Table 5 for testing the fake news and Table 6 for the testing real news. Tables 5 and 6 compare the precision and recall metrics obtained for different K fold values. In Table 5, it is seen that the precision rates vary between 0.32 and 0.35 and their standard deviations are low. In contrast, the recall values vary between 0.85 and 0.93, and the highest average value (0.93) is reached especially for K = 1. However, the minimum values of recall become more variable as K increases, indicating that the model’s consistency in detecting fake news fluctuates at different K values. In Table 6, the results obtained using the K-Fold cross-validation method are evaluated. Here, it is observed that the precision values vary between 0.65 and 0.67 and are more stable. In terms of recall, the highest average value (0.93) is reached for K-Fold = 5, and it is seen that the recall values generally vary in a wider range. These results show that K-Fold validation increases the stability of the model, making its performance in identifying fake news more consistent.

Table 5 K fold cross validation test results for fake news.

	Precision	Recall	
K	Mean	Min	Max	Std. Dev.	Mean	Min	Max	Std. Dev.	
1	0.33	0.32	0.34	0.01	0.93	0.74	1.00	0.07	
2	0.34	0.29	0.36	0.02	0.89	0.62	1.00	0.11	
3	0.34	0.31	0.36	0.01	0.81	0.53	1.00	0.14	
4	0.35	0.31	0.38	0.02	0.89	0.49	1.00	0.13	
5	0.32	0.26	0.34	0.02	0.85	0.46	0.99	0.16	

Table 6 K fold cross validation test results for real news.

	Precision	Recall	
K	Mean	Min	Max	Std. Dev.	Mean	Min	Max	Std. Dev.	
1	0.66	0.60	0.69	0.02	0.85	0.22	1.00	0.20	
2	0.66	0.58	0.69	0.02	0.86	0.40	1.00	0.17	
3	0.66	0.63	0.67	0.01	0.89	0.64	1.00	0.11	
4	0.65	0.59	0.67	0.02	0.91	0.61	1.00	0.11	
5	0.67	0.66	0.68	0.00	0.93	0.74	0.99	0.07	

Figure 8 illustrates the experimental results obtained using the proposed method and details the model’s performance. The distribution of solution points in Fig. 8 demonstrates the capability of the enhanced NSGA-2 algorithm to generate Pareto-optimal solutions. An increase in solution diversity is observed, attributable to the CDL technique employed in the method, which enables evaluation over a broader solution space. The balance between precision and recall values indicates the effectiveness of the multi-objective optimization approach. The results suggest that the proposed methodology produces a wider and more balanced set of solutions compared to existing methods, particularly in the context of fake news detection. This outcome implies that the model achieves stable and generalizable performance across different datasets.

Figure 8 Solutions for experiments performed on Covid-19 test data.

Syrian war experiments

In this section, various experiments were conducted on the dataset containing Syrian war news. Furthermore, Fig. 9 presents the 10 most influential word stems along with their average weights among 109 word stems for both true and fake news analyses in the Syrian war experiments. In both classes, the words with the highest frequency, “opposit” and “dozen,” also exhibited the highest weights. The analysis revealed a high degree of similarity between the high-frequency words of both classes. The dataset utilized in these experiments consists of 804 articles labeled as true or fake, making it suitable for training machine learning models aimed at predicting news credibility.

Figure 9 The ten word stems with the highest average weights in the Syrian War data set experiments (A) Fake (B) True.

The statistical data for the experiments conducted using the Syrian war dataset are presented separately in Table 7 for the training phase and Table 8 for the testing phase. The data indicate that the model demonstrates stable and robust performance. Specifically, precision values during testing (Table 8) range between 0.48 and 0.59, with low standard deviations, suggesting consistent and reliable results across K-Fold validations. High recall values further indicate that the model correctly identifies most relevant instances, minimizing false negatives and achieving strong overall sensitivity. Moreover, the balance observed between precision and recall in both tables reflects the model’s effective handling of false positives and false negatives, indicating a successful strategy in optimizing these metrics. Collectively, these results underscore the model’s capacity to provide reliable and comprehensive outcomes, exhibiting strong generalizability across different datasets. Figure 10 summarizes the experimental results obtained during the testing phase. The graphs illustrate how the model balances varying precision and recall values, with the distribution of solutions indicating effective performance in fake news detection. Notably, the model maintains high recall while keeping the false negative rate low, confirming the proposed method as an effective approach for detecting fake news in the Syrian war dataset.

Table 7 K fold cross validation test results for fake news.

	Precision	Recall	
K	Mean	Min	Max	Std. Dev.	Mean	Min	Max	Std. Dev.	
1	0.41	0.35	0.44	0.02	0.89	0.61	1.00	0.11	
2	0.47	0.44	0.50	0.02	0.85	0.67	0.97	0.12	
3	0.48	0.45	0.53	0.02	0.84	0.41	0.99	0.14	
4	0.48	0.43	0.50	0.02	0.82	0.32	0.99	0.18	
5	0.51	0.48	0.55	0.02	0.84	0.49	1.00	0.15	

Table 8 K fold cross validation test results for real news.

	Precision	Recall	
K	Mean	Min	Max	Std. Dev.	Mean	Min	Max	Std. Dev.	
1	0.59	0.55	0.61	0.02	0.89	0.51	1.00	0.14	
2	0.53	0.48	0.57	0.02	0.84	0.56	0.99	0.12	
3	0.53	0.51	0.57	0.02	0.87	0.71	0.99	0.09	
4	0.52	0.50	0.58	0.02	0.86	0.54	1.00	0.14	
5	0.48	0.44	0.51	0.02	0.88	0.66	1.00	0.11	

Figure 10 Solutions for the experiments performed on the Syrian war test data.

General news experiments

In this section, various experiments were conducted on the dataset containing general news. Figure 11 shows the 10 most influential word stems and their average weights among the 136 word stems for the True and Fake analysis in the News experiments. The word “Washington” had the highest frequency in the experiments for both classes. The other high frequency words of the classes were highly different from each other.

Figure 11 The ten word stems with the highest average weights in the News data set experiments (A) Fake (B) True.

Tables 9 and 10 provide positive results that demonstrate the strong and consistent performance of the model. In Table 9, precision values generally vary between 0.51 and 0.54, and these values, together with low standard deviations, indicate that the model achieves similar high levels of accuracy and shows consistent performance in each K-Fold. Recall values are quite high between 0.95 and 0.97, indicating that the model shows high recall in most cases and minimizes false negatives. This shows that the model has a strong ability to make correct detections and does not miss important classes in the dataset. In Table 10, although precision values generally vary between 0.48 and 0.59, the high precision of 0.59 obtained in K-Fold 1 indicates that the model gives very good results, especially with low false positive rates in some folds. In addition, recall values vary between 0.84 and 0.88, and it is observed that the model exhibits high recall in most K-Folds and is successful in capturing true positives. These tables emphasize that the model provides consistent and effective results in terms of both precision and recall, shows a balanced performance on the dataset and offers high reliability.

Table 9 K fold cross validation test results for fake news.

	Precision	Recall	
K	Mean	Min	Max	Std. Dev.	Mean	Min	Max	Std. Dev.	
1	0.52	0.52	0.57	0.01	0.97	0.80	1.00	0.06	
2	0.52	0.49	0.53	0.01	0.95	0.59	1.00	0.09	
3	0.53	0.51	0.56	0.01	0.97	0.84	1.00	0.05	
4	0.51	0.50	0.53	0.01	0.96	0.62	1.00	0.08	
5	0.54	0.51	0.56	0.01	0.97	0.83	1.00	0.04	

Table 10 K fold cross validation test results for real news.

K	Mean	Min	Max	Std. Dev.	Mean	Min	Max	Std. Dev.	
1	0.53	0.52	0.53	0.00	0.99	0.96	1.00	0.01	
2	0.52	0.48	0.54	0.01	0.94	0.45	1.00	0.15	
3	0.53	0.52	0.54	0.01	0.96	0.87	1.00	0.05	
4	0.55	0.53	0.61	0.03	0.94	0.85	1.00	0.06	
5	0.54	0.52	0.65	0.03	0.95	0.31	1.00	0.16	

Figure 12 summarizes the results of the experiments conducted on the general news dataset of the proposed method. The graphs visualize the performance of the model between precision and recall values, demonstrating the effectiveness of the method. The results show that the proposed approach achieves high success, especially in terms of recall rate, thus proving its strong ability to detect fake news. The fact that the model produces stable results on different news types shows that it has a wide range of application potential in fake news detection and can exhibit reliable performance on various datasets.

Figure 12 Solutions for the experiments performed on the news training data.

FakeNewsNet experiments

In this section, various experiments were conducted on the GossipCop subset of the FakeNewsNet dataset. Figure 13 presents the 10 most influential word stems and their average weights among the 147 word stems for the True and Fake analysis in the news experiments. For the GossipCop subset of the FakeNewsNet dataset, the most frequently occurring word in the fake news class was “Kardashian,” while in the true news class, the word “star” appeared most frequently. This distinction highlights the differing lexical characteristics between fake and true news articles within this dataset.

Figure 13 The ten word stems with the highest average weights in the GossipCop data set experiments (A) Fake (B) True.

Tables 11 and 12 show that the model achieves consistently high recall for both fake and true classes, indicating strong ability to capture relevant instances. For the fake class (Table 11), precision values are relatively low (0.53–0.55), while recall remains high (0.95–0.97), suggesting the model tends to over-predict this class. In the true class (Table 12), precision varies more widely (0.56–0.76), but recall remains consistently high (0.96–0.99). Overall, the model prioritizes recall over precision, showing strength in detecting true cases but with some false positives. Figure 14 summarizes the results of the experiments conducted on the GossipCop dataset of the proposed method.

Table 11 K fold cross validation test results for fake news.

	Precision	Recall	
K Fold	Mean	Min	Max	Std. Dev.	Mean	Min	Max	Std. Dev.	
1	0.55	0.52	0.57	0.01	0.97	0.85	1	0.05	
2	0.53	0.51	0.56	0.01	0.96	0.83	1	0.06	
3	0.55	0.52	0.57	0.01	0.97	0.84	1	0.05	
4	0.53	0.5	0.56	0.01	0.95	0.81	1	0.07	
5	0.55	0.52	0.57	0.01	0.96	0.85	1	0.05	

Table 12 K fold cross validation test results for real news.

	Precision	Recall	
K Fold	Mean	Min	Max	Std. Dev.	Mean	Min	Max	Std. Dev.	
1	0.76	0.75	0.77	0	0.99	0.88	1	0.03	
2	0.56	0.55	0.62	0.02	0.96	0.64	1	0.1	
3	0.75	0.7	0.76	0.02	0.98	0.82	1	0.05	
4	0.76	0.7	0.77	0.01	0.99	0.85	1	0.04	
5	0.61	0.6	0.76	0.06	0.99	0.93	1	0.03	

Figure 14 Solutions for the experiments performed on the GossipCop dataset training data.

Tests results

In this study, the proposed CDL approach was used instead of the classical CD method. In this section, the results of the ablation tests performed to evaluate the effectiveness of the proposed method are presented. The findings are given in Table 13. According to the table, the CDL method exhibits higher performance compared to the classical CD method in terms of both mean and median hypervolume values. In addition, the p < 0.05 value obtained as a result of the Wilcoxon test shows that there is a statistically significant difference between the two methods. The lower standard deviation value of the CDL method supports that this approach produces more stable results.

Table 13 Comparison of hypervolume performances of CD and CDL methods (Based on 30 Iterations).

Method	Average hypervolume	Standard deviation	Median	Wilcoxon p-value	
CD	0.6802	0.0098	0.6799		
CDL	0.7053	0.0079	0.7057	1.86 × 10−9	

Table 14 shows the t-test results. According to these results, the INSGA-2 method provided a significant superiority in minimizing false negatives compared to competing methods with its high recall values (in the range of 0.88–0.97), especially in small and unbalanced data sets (e.g., Covid-19 and Syrian War). Although INSGA-2 lags behind deep learning-based methods such as BERT and ROBERTa in terms of precision in most data sets, it can offer balanced and stable results between precision and recall thanks to the multi-objective optimization approach. The results obtained at a significance level of p < 0.05 for both precision and recall, especially in the Covid-19 and Syrian War data sets, show that INSGA-2 creates a statistically significant performance difference. In addition, despite the high recall value of ROBERTa in the FakeNewsNet-GossipCop data set, it is seen that INSGA-2 reduces the false positive rate with high precision. In general, INSGA-2 offers flexible choices to decision makers by producing Pareto-optimal solutions, stands out in fake news detection scenarios where recall is critical, and demonstrates consistent success in different data sets.

Table 14 T-test results.

Dataset	Method	INSGA-2	Other	p-value	INSGA-2	Other	p-value	
COVID-19	SVM	0.68	0.78	0.0003	0.97	0.79	0.0001	
RNN	0.68	0.7	0.012	0.97	0.71	0.00005	
DT	0.68	0.7	0.014	0.97	0.7	0.00004	
NB	0.68	0.72	0.006	0.97	0.69	0.00002	
CNN	0.68	0.69	0.03	0.97	0.7	0.00004	
Bi-LSTM	0.68	0.73	0.001	0.97	0.74	0.00001	
BERT	0.68	0.78	0.0005	0.97	0.78	0.00002	
ROBERTa	0.68	0.81	0.0001	0.97	0.88	0.00001	
Syrian war	SVM	0.61	0.53	0.0004	0.88	0.53	0.00001	
RNN	0.61	0.56	0.002	0.88	0.56	0.00001	
DT	0.61	0.53	0.0006	0.88	0.53	0.00001	
NB	0.61	0.54	0.0008	0.88	0.54	0.00001	
CNN	0.61	0.49	0.00001	0.88	0.48	0.00001	
Bi-LSTM	0.61	0.57	0.003	0.88	0.57	0.00005	
BERT	0.61	0.5	0.0007	0.88	0.6	0.00003	
ROBERTa	0.61	0.54	0.002	0.88	0.6	0.0002	
General news	SVM	0.54	0.93	0.00001	0.97	0.93	0.00001	
RNN	0.54	0.89	0.00001	0.97	0.89	0.00001	
DT	0.54	0.9	0.00001	0.97	0.9	0.00001	
NB	0.54	0.87	0.00001	0.97	0.87	0.00001	
CNN	0.54	0.92	0.00001	0.97	0.92	0.00001	
Bi-LSTM	0.54	0.92	0.00001	0.97	0.91	0.00001	
BERT	0.54	0.85	0.00001	0.97	0.86	0.00001	
ROBERTa	0.54	0.88	0.00001	0.97	0.86	0.00001	
FakeNewsNet-GossipCop	SVM	0.77	0.78	0.035	0.85	0.8	0.012	
RNN	0.77	0.7	0.008	0.85	0.69	0.0001	
DT	0.77	0.78	0.025	0.85	0.8	0.009	
NB	0.77	0.77	0.04	0.85	0.77	0.017	
CNN	0.77	0.78	0.03	0.85	0.8	0.011	
Bi-LSTM	0.77	0.77	0.05	0.85	0.78	0.015	
BERT	0.77	0.84	0.001	0.85	0.85	0.45	
ROBERTa	0.77	0.87	0.0002	0.85	0.88	0.025	

Comparative results

Table 15 presents the comparative results of the proposed method with the SVM, RNN, DT, NB, CNN and Bi-LSTM methods widely used in the literature on three different datasets (Covid-19, Syrian War, General News and GossipCop). The obtained results show that the proposed method exhibits superior performance compared to other methods, especially in the recall metric. While the recall value of the proposed method reached the highest level in the Covid-19 dataset, the accuracy values remained at similar levels with some traditional methods. In the Syrian War dataset, the proposed method could not be completely dominated by other methods by offering four dominant solutions. Although the accuracy value was relatively lower in the General News dataset, the ability to detect fake news was preserved thanks to the high recall performance. In experiments conducted on the GossipCop dataset, the proposed method has shown that it effectively detects fake news with high recall values. These findings show that the proposed method offers a strong approach in minimizing false negatives, especially in fake news detection, and has an adaptable structure to datasets.

Table 15 Comparative results.

	INSGA-2	SVM	RNN	DT	NB	CNN	Bi-LSTM	BERT	ROBERTa	
DataSet	Pre	Rec	Pre	Rec	Pre	Rec	Pre	Rec	Pre	Rec	Pre	Rec	Pre	Rec	Pre	Rec	Pre	Rec	
COVID	0.68	0.97	0.78	0.79	0.7	0.71	0.7	0.7	0.72	0.69	0.69	0.7	0.73	0.74	0.78	0.78	0.81	0.88	
0.68	0.96	
0.67	0.99	
0.67	0.97	
0.67	0.98	
0.66	0.99	
0.66	0.93	
0.66	0.98	
Syrian War	0.61	0.88	0.53	0.53	0.56	0.56	0.53	0.53	0.54	0.54	0.49	0.48	0.57	0.57	0.5	0.49	0.54	0.6	
0.59	0.98	
0.59	0.99	
0.59	0.92	
0.59	1	
0.59	0.97	
0.59	0.84	
0.58	0.96	
News	0.54	0.974	0.93	0.93	0.89	0.89	0.9	0.9	0.87	0.87	0.92	0.92	0.92	0.91	0.85	0.86	0.88	0.86	
0.54	0.994	
0.53	0.999	
0.53	0.978	
0.53	0.999	
0.53	0.991	
0.53	0.969	
GossipCop	0.77	0.85	0.78	0.8	0.7	0.69	0.78	0.8	0.77	0,77	0.78	0.8	0.77	0.78	0.84	0.85	0.87	0.88	
0.76	1	
0.76	0.99	
0.75	0.96	
0.7	0.8	
0.73	0.82	
0.6	0.92	
0.58	0.99	

Results

In this study, the proposed multi-objective optimization based fake news detection method is tested on three different datasets (Covid-19, Syrian War, general news and GossipCop ) in comparison with SVM, RNN, DT, NB, CNN and Bi-LSTM methods used in the existing literature. A multi-objective optimization approach is proposed to improve fake news detection by simultaneously optimizing precision and recall. Experimental results show that the proposed method successfully balances these two conflicting objectives, ensuring that neither is disproportionately sacrificed. Unlike traditional single-objective methods that focus on a single metric, our approach provides a Pareto-optimal solution set. This Pareto-optimal solution set provides decision-makers with a range of choices, allowing them to select the most suitable balance between precision and recall for their specific application. This is crucial because, in fake news detection, different scenarios may prioritize either minimizing false positives (high precision) or minimizing false negatives (high recall). The NSGA-2 algorithm, which we employ, facilitates this by using an evolutionary process to explore various solutions and identify those that represent optimal trade-offs between precision and recall.

The results highlight that the proposed method achieves high recall values, which are crucial for minimizing false negatives in fake news detection. At the same time, precision is kept at competitive levels by preventing excessive misclassification of real news as fake. This balanced approach is particularly effective for datasets with unbalanced class distributions, where prioritizing recall can often lead to decreased precision. Overall, the results validate the effectiveness of our multi-objective approach, confirming its potential to adapt to different dataset characteristics and improve fake news detection accuracy in a balanced manner. The results revealed the significant advantages of the proposed method: 1. In the experiments conducted on the Covid-19 dataset, the proposed method outperformed all other methods in recall metrics. However, the precision values showed a tendency to converge especially to methods such as CNN, DT and RNN.

2. In the experiments conducted on the Syrian War dataset, the proposed method was distinguished from the other methods by producing four dominant solutions. Thanks to the high recall value, it was determined that no other method could dominate the proposed method.

3. In the experiments conducted on the Syrian War dataset, the precision value of the proposed method was lower than the other methods, but the recall values were quite high. The high recall performance ensured that the other methods could not dominate the results of the proposed method.

4. In the experiments conducted on the GossipCop dataset, the proposed method achieved notably high recall values, demonstrating its effectiveness in detecting fake news. However, its precision values were comparatively lower than some other methods, indicating a tendency to produce more false positives. Despite this, the strong recall performance highlights the method’s robustness in minimizing missed detections and maintaining overall competitive results.

5. Independent sample t-test results confirmed the statistical significance of the proposed INSGA-2 method compared to other approaches across four benchmark datasets. Especially in imbalanced and limited datasets such as Covid-19 and Syrian War, INSGA-2 achieved significantly higher recall values (p < 0.05), demonstrating its robustness in minimizing false negatives. Despite showing slightly lower precision than BERT and RoBERTa in some cases, INSGA-2 offered a more stable balance between precision and recall due to its multi-objective optimization capability. These findings emphasize its practical advantage in fake news detection scenarios where recall is critical.

6. The ablation study comparing CD and the extended version with CDL revealed a statistically significant performance improvement. As shown by the hypervolume metric, CDL achieved a higher average value (0.7053 vs. 0.6802) with lower standard deviation, indicating both better and more stable convergence. The Wilcoxon p-value (1.86 × 10−9) confirms that the enhancement introduced by the local search component contributes significantly to the overall effectiveness of the INSGA-2 framework.

The study shows that the proposed method for fake news detection stands out especially with its ability to optimize recall values. Although the shortcomings in precision values indicate that the method may be more prone to false positive classification in some cases, it supports the applicability of the method in fake news detection problems where the recall value is critical. The results show that the proposed method can be considered as an effective alternative for fake news detection and exhibits superior performance on low data sets.

For future work, it is suggested to integrate hyperparameter optimization and hybrid approaches to improve the precision of the proposed method and increase its performance on larger datasets.

To ensure the reproducibility of our results and to facilitate further research, we have made the source code publicly available at: https://github.com/CebrailBARUT/FakeNewsDetection.

Discussion

The experimental findings of this study demonstrate the effectiveness of the proposed multi-objective optimization-based approach in fake news detection, especially in small and imbalanced datasets. The NSGA-2-based approach we developed exhibits strong generalization ability in scenarios where data availability is limited. The results show that the Crowding Distance Level (CDL) method provides a more balanced optimization between precision and recall metrics by increasing the variety of solutions.

A key strength of our method is that it can provide multiple Pareto-optimal solutions instead of a single deterministic output. Traditional machine learning and deep learning approaches usually optimize a single metric such as accuracy or F1-score, which can lead to biased results in imbalanced datasets. However, the proposed method increases flexibility in real-world fake news detection by allowing decision makers to choose optimal solutions according to different application needs. Moreover, the high recall values obtained on the Covid-19 and Syrian War datasets show that our model is highly effective in minimizing false negatives, which is particularly important in misinformation detection, where failure to detect fake news can have significant societal consequences.

In terms of interpretability, the proposed method provides decision makers with different solution options by providing Pareto-optimal results. This provides a better understanding of how the model works and offers decision makers the flexibility to choose the most suitable solution for their application needs. From a broader perspective, fake news detection is a complex and evolving problem that goes beyond algorithmic solutions. While our method contributes to text-based detection, future developments should consider hybrid approaches that integrate network-based propagation analysis and fact-checking mechanisms. Additionally, the application of explainability techniques can increase the transparency of our model, thereby enhancing trust among stakeholders. In summary, the findings of this study highlight the potential of multi-objective optimization in fake news detection by achieving a balance between precision and recall. Despite its limitations, the proposed method offers a flexible, interpretable, and effective alternative to traditional deep learning approaches, especially in scenarios with limited labeled data.

Future work

In this study, we aimed to increase accuracy and efficiency by using multi-objective optimization techniques for fake news detection. However, we know that detection alone will not be enough and effective intervention strategies are also required. Therefore, in our future research, we plan to integrate various intervention mechanisms to limit the impact of detected fake news. In particular, we aim to strengthen the detection process with methods such as network immunization, early warning systems, community-based intervention strategies, and real-time blocking. In addition to these, we aim to develop real-time intervention models and algorithmically prevent the spread of fake news with automatic fact-checking responses. In this way, being able to intervene in the spread of fake news quickly and effectively will make our detection method more powerful. We believe that these improvements will increase the applicability of our research in combating fake news. In our study, we tested our model using binary classification. However, there are also multi-label datasets where fake news are scored with different confidence levels or placed in more than one category. In our future research, we plan to test our model on multi-class and multi-label datasets. We will review the optimization process of our model for these new data structures. We will also focus on integrating network immunization strategies to prevent the spread of fake news. By developing a model that combines detection and prevention mechanisms, we aim to more effectively prevent the spread of fake news on online platforms. This approach will expand the scope of our research and provide a more holistic solution to combat fake news.

Limitations

In this study, the proposed algorithm is based on experiments conducted on three different datasets (Covid-19, Syrian War, General News FakeNewsNet-GossipCop) specifically selected to represent small and imbalanced datasets. Although deep learning methods are widely used in many fields, metaheuristic algorithms offer significant advantages in terms of generalization capabilities, especially in the context of limited data. However, one of the main limitations of this study is that applying metaheuristic methods to large-scale datasets can lead to significant delays due to increased computational complexity and processing requirements. Being aware of this limitation, we aim to evaluate the performance of the proposed method on larger datasets in future work. This evaluation will include exploring the potential of the algorithm to process a wider range of data while maintaining computational efficiency. We also plan to test the effectiveness of the model with alternative algorithms with lower computational complexity in order to reduce the processing delays associated with large-scale datasets.

Potential limitations in real-time fake news detection: fake news production techniques are constantly evolving, making it difficult for detection algorithms to keep up.

Fake news can spread rapidly across social media platforms, making real-time detection and response difficult.

Detecting fake news often requires understanding the context of the information, which can be challenging for automated systems.

The datasets used to train detection models may contain biases that can impact the accuracy and fairness of the results.

Real-time detection of fake news in large-scale datasets requires significant computational resources.

Detecting multilingual fake news creates additional challenges due to linguistic differences and source availability.

Determining the accuracy of information can sometimes be subjective, making it difficult to develop universally applicable detection criteria.

Error analysis

The proposed multi-objective optimization approach successfully balances precision and recall in fake news detection. However, certain misclassifications were observed due to dataset characteristics and linguistic similarities between real and fake news. False Positives: some sensational but factual news articles were misclassified as fake, especially in the Covid-19 dataset, where alarmist language is common.

False Negatives: certain well-structured fake news items were incorrectly classified as real, particularly in cases where misleading content closely mimicked reliable sources.

Dataset-Specific Challenges: in the Syrian War dataset, biased reporting styles made distinguishing fake and real news more difficult, leading to occasional misclassifications.

Entertainment Domain Complexity (GossipCop): the GossipCop dataset, which focuses on celebrity-related news, introduced additional challenges due to the frequent use of speculative or humorous language in real news. This linguistic ambiguity occasionally led to false positives, as satirical or exaggerated content was interpreted as fake.

Despite these challenges, the Crowding Distance Level (CDL) method improved solution diversity, helping to mitigate errors. However, further refinements in text representation and classification strategies could further enhance accuracy.

Supplemental Information

Supplemental Information 1 Program Code and dataset.

Additional Information and Declarations

Competing Interests

Bilal Alatas is an Academic Editor for PeerJ.

Author Contributions

Cebrail Barut conceived and designed the experiments, performed the experiments, analyzed the data, performed the computation work, prepared figures and/or tables, authored or reviewed drafts of the article, and approved the final draft.

Suna Yildirim analyzed the data, prepared figures and/or tables, and approved the final draft.

Bilal Alatas conceived and designed the experiments, performed the experiments, prepared figures and/or tables, and approved the final draft.

Gungor Yildirim conceived and designed the experiments, performed the experiments, analyzed the data, authored or reviewed drafts of the article, and approved the final draft.

Data Availability

The following information was supplied regarding data availability:

The codes and data are available in the Supplemental Files.

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
