# Peer review of "Innovative multi objective optimization based automatic fake news detection"

_PeerJ Computer Science, doi:10.7717/peerj-cs.3016_

## Round 0.1 · original submission · Major Revisions

There are a variety of improvements needed, as captured by a wide range of issues identified by the reviewers. In terms of presentation, you need to have a well structured and reasonably exhaustive survey fo related works, with which you should also explain the novelties of your approach, and later on in experiments, use the state-of-the-art to benchmark the performance of your approach, considering not only the quality of results, but also the scalability of the approach in terms of run time, and with larger and wider variety of data sets. Designing experiments and measuing the results in a manner which facilitates better explainability/interpretability of the results are also expected.

Reviewer 1 ·

Basic reporting

The article presents an approach for fake news detection.
Due to the datasets, the problem is boiled down to binary classification.
The novelty is very limited, as many research works present actual deep learning models that use multimodal techniques, e.g., DANES (https://doi.org/10.1016/j.knosys.2024.111715), GETAE (https://arxiv.org/abs/2412.01825), MisRoBÆRTa (https://doi.org/10.3390/math10040569), or the classical machine learning algorithms employed in this study on multiple datasets (https://doi.org/10.3390/math11030508).

1. There is no Related Work section.

2. The article lacks a lot of current literature on fake news detection and mitigation and the related work must be improved as follows (please respond individually to each point in the answer to reviewers):
a. In the current literature, many works are proposing the use of word embeddings [1], transformers [2], sentence transformers [3], and document embeddings [4], a mixture of experts [5] for detecting fake/harmful information. How does the proposed architecture compare to these methods?
b. The current literature discusses how ensemble methods that incorporate information diffusion [6] or social network information [7] can improve the detection task. How does the proposed architecture compare to these methods?
c. The current literature contains a large volume of work dedicated to network immunization after performing fake news detection, such as proactive approaches [8], tree-based approaches [9], community-based approaches [10], or real-time approaches [11] and architectures [12]. What are the future directions of this work? Detection for the sake of detection is pointless.

3. The text preprocessing methods are not new and are widely used by different architectures [13]. Please cite accordingly.

4. The weight used for a word is known in the literature as the double normalized K Term Frequency with K=0. Please use the correct terminology and cite accordingly [14].


5. Why use TF to weight words? Why not word embeddings [1], transformers embeddings [2], sentence transformers [3], and document embeddings [4], a mixture of experts [5]?

6. The evaluation metrics should be chosen carefully depending on the dataset used [15] (balanced vs imbalanced).

7. The study is missing a clear exploratory data analysis for the datasets.

8. The algorithms are widely used in the literature. What is new?

9. The experiments should be improved using the proper machine/deep learning evaluation methods (please respond individually to each point in the answer to reviewers)
a. There is no k-fold cross-validation. Splitting the date only on the train and test set is not cross-validation. In the experiment, provide the mean and standard deviation computed after applying k-fold cross-validation
b. There is no time performance evaluation.
c. The experiments are done on a tiny dataset. How does the proposed model perform on a larger dataset?
d. The experiments are done on 2 binary datasets What happens when using datasets with multiple labels?
e. The evaluation metrics should be appropriately selected for the type of dataset used [15] (balanced vs imbalanced).
f. There is no comparison with state-of-the-art models.
g. There is no ablation testing for the deep learning models.
h. The experiments do not show if the proposed models generalize well. Experiments on larger datasets are required.
i. Why are the figures missing from the manuscript? Please add them in the revised version.

10. A discussion section is missing. Did we learn nothing from this work?

11. A limitations section is missing. It seems that using this model we have solved the fake news problem.

12. For future work, maybe the authors should also think about integrating network immunization strategies to stop the spread of fake news online [8,9,10] and develop full architectures for fake news detection and mitigation [11,12].

13.  For reproducibility purposes, the authors should make the code publicly available. Without publicly available source code, models, and datasets, a shadow of doubt can fall on the results of this work. How can anyone check if these results are real or made up without the code, models, and datasets? I have seen many papers in my academic life where the authors say they provide the code and models upon request after publication and ignore emails afterwards. Thus, I ask the authors to share the code and models using a public license on a public code-sharing platform, e.g., GitHub. Without this requirement, I am inclined to reject the manuscript.

14. Please do a proofreading and spelling check before resubmission.


[1] https://scholar.google.com/scholar?q=word+embeddings+misinformation+detection
[2] https://scholar.google.com/scholar?q=transformers+misinformation
[3] https://scholar.google.com/scholar?q=fake+news+sentence+transformers
[4] https://scholar.google.com/scholar?q=fake+news+document+embeddings
[5] https://scholar.google.com/scholar?q=+language-based+mixture+of+transformers
[6] https://scholar.google.com/scholar?q=graph+information+enhanced+ensemble+architecture+fake+news+detection
[7] https://scholar.google.com/scholar?q=deep+neural+network+ensemble+architecture+fake+news+detection
[8] https://scholar.google.com/scholar?q=social+network+immunization+harmful+speech
[9] https://scholar.google.com/scholar?q=real+time+social+media+tree+algorithm+mitigation
[10] https://scholar.google.com/scholar?q=network+immunization+community+detection+fake+news+detection
[11] https://scholar.google.com/scholar?q=distributed+system+misinformation+detection+community+detection
[12] https://scholar.google.com/scholar?q=harmful+content+detection+and+mitigation+social+media+platforms
[13] https://scholar.google.com/scholar?q=scalable+document-based+architecture+text+analysis
[14] https://scholar.google.com/scholar?q=benchmarking+keyword+processing
[15] https://scholar.google.com/scholar?q=classification+imbalanced+data+sets+decision+trees

Experimental design

1. The study is missing a clear exploratory data analysis for the datasets.

2. The algorithms are widely used in the literature. What is new?

3. The experiments should be improved using the proper machine/deep learning evaluation methods (please respond individually to each point in the answer to reviewers)
a. There is no k-fold cross-validation. Splitting the date only on the train and test set is not cross-validation. In the experiment, provide the mean and standard deviation computed after applying k-fold cross-validation
b. There is no time performance evaluation.
c. The experiments are done on a tiny dataset. How does the proposed model perform on a larger dataset?
d. The experiments are done on 2 binary datasets What happens when using datasets with multiple labels?
e. The evaluation metrics should be appropriately selected for the type of dataset used [15] (balanced vs imbalanced).
f. There is no comparison with state-of-the-art models.
g. There is no ablation testing for the deep learning models.
h. The experiments do not show if the proposed models generalize well. Experiments on larger datasets are required.
i. Why are the figures missing from the manuscript? Please add them in the revised version.

Validity of the findings

Employ proper machine/deep learning evaluation methods (see Experimental design comments)

For reproducibility purposes, the authors should make the code publicly available. Without publicly available source code, models, and datasets, a shadow of doubt can fall on the results of this work. How can anyone check if these results are real or made up without the code, models, and datasets? I have seen many papers in my academic life where the authors say they provide the code and models upon request after publication and ignore emails afterwards. Thus, I ask the authors to share the code and models using a public license on a public code-sharing platform, e.g., GitHub. Without this requirement, I am inclined to reject the manuscript.

Cite this review as

·

Basic reporting

The authors have used a metaheuristic approach for the classification of fake news.
The concept is new, and the usage of crowd distancing level method is novel in this domain.

1. The authors need to update the paper and explain the multi-objective in the results section more clearly. The results in Table 1 Comparative results do not reflect clearly that Precision and Recall both have been raised correctly. Overall improvement in very high Recall can also be tuned in other strategies.

2. There is no figure for the proposed framework/or pseudocode. Please add.

3. Authors say, “While deep learning based approaches produce successful results on large and complex data sets, their performance on small data sets is often limited.” – Please provide a reference for such statements

4. Authors say, “metaheuristics can produce faster results compared to deep learning based solutions.” – Please provide a reference as its not true for many problem statements. Metaheuristics may be faster on optimization problems but not on pattern matching.

5. There is no limitation and error analysis section. Were there no errors reported?

Experimental design

1. On some datasets, the Recall of 1.0 indicates overfitting. What steps were taken by the authors to mitigate the overfitting?

2. Please provide the hyperparameters for the ML models used. What hyperparameters were employed for SVM, RNN, DT, NB, CNN, and Bi-LSTM?

3. How is the interpretability of the results? Please clarify more clearly.

Validity of the findings

1. Please compare your results with some SOTA research papers in deep learning and Machine Learning that have used the same datasets as yours.

Reviewer 3 ·

Basic reporting

This study proposes a multi-objective optimization-based approach for fake news detection, leveraging an improved version of the Non-Dominated Sorting Genetic Algorithm-2 (NSGA-2). Unlike traditional fake news detection models that optimize a single metric, this research focuses on simultaneous optimization of precision and recall, leading to more balanced classification outcomes.
The paper introduces a new Crowding Distance Level (CDL) method, replacing the standard Crowding Distance (CD) method in NSGA-2, which enhances the diversity of Pareto-optimal solutions. The proposed method is tested on three real-world datasets (Covid-19, Syrian war, and general news) and compared against widely used machine learning models such as SVM, RNN, Decision Trees, Naïve Bayes, CNN, and Bi-LSTM.

Experimental design

• The paper does not provide a runtime comparison of the proposed method against deep learning models.
• Metaheuristic algorithms (like NSGA-2) can be computationally expensive, especially when applied to large datasets.
Suggested Improvement:
• Conduct a runtime analysis to compare execution time with deep learning models.
• Discuss the scalability of the proposed method when dealing with large datasets (e.g., millions of records).

• While the study identifies important word stems (e.g., "Covid", "isolation", "opposition"), there is no discussion on their semantic relevance.
• Fake news detection models must not only be accurate but also explainable, especially for use in journalism and policymaking.
Suggested Improvement:
• Provide a qualitative analysis of the top-ranked word stems.
• Discuss the practical implications of selected features (e.g., how they help distinguish fake vs. real news).
• Use SHAP (Shapley Additive Explanations) or LIME (Local Interpretable Model-agnostic Explanations) for interpretability.
• The paper does not explore how changes in NSGA-2 parameters (e.g., population size, crossover/mutation rates) affect model performance.
• Different problem domains may require different hyperparameter settings, making generalizability a concern.
Suggested Improvement:
• Conduct a hyperparameter sensitivity analysis to determine optimal parameter ranges.
• Test automated hyperparameter tuning methods (e.g., Bayesian Optimization, Grid Search).

Validity of the findings

• Metaheuristic-based methods can overfit small datasets by optimizing too aggressively for specific data distributions.
• The paper does not explicitly discuss how overfitting was mitigated.
Suggested Improvement:
• Perform k-fold cross-validation to ensure generalizability.
• Implement regularization techniques to prevent overfitting.
• Discuss potential limitations in real-time fake news detection.


• The method works well on small datasets, but its scalability to large-scale social media data remains unclear.
• Metaheuristic-based methods are generally slower than deep learning models when handling millions of records.

Additional comments

• The paper does not discuss real-world integration challenges (e.g., deployment in news verification systems, fact-checking organizations).
• Real-time fake news detection requires models to process data at high speeds, but the paper does not evaluate this aspect.

• Fake news datasets often suffer from class imbalance (i.e., more real news than fake news).
• The paper does not explore how class imbalance affects optimization performance.

• Implement parallel computing (e.g., GPU acceleration) to handle large datasets.
• Conduct Big-O complexity analysis to estimate performance scalability.

• Use SHAP/LIME to explain why specific words contribute to classification.
• Conduct case studies on real-world fake news to validate model predictions.

• Test automated tuning methods (e.g., Bayesian Optimization).
• Conduct an ablation study to determine the effect of each parameter.

• Implement cross-validation to validate generalizability.
• Use data augmentation techniques to handle imbalanced datasets.

Cite this review as

Reviewer 4 ·

Basic reporting

Paper needs more work and resubmission -- here are the main points at this stage:


- Clearly define the concept of metaheuristics and provide a proper citation to support the claim that metaheuristics outperform deep learning on small datasets.

- While the paper states that three datasets were used for the experiments, it does not include details on how the data was preprocessed. Additionally, there is no clear explanation regarding the dataset characteristics, such as whether it is balanced or imbalanced and its modality.

- The novelty of this approach compared to prior applications of NSGA-2 in fake news detection needs to be explicitly highlighted.

- In the methodology section, provide a justification for selecting the Crowding Distance Level (CDL) method. Additionally, explain the rationale behind the parameter settings and elaborate on the feature selection process.

- The authors should ensure the availability of code to facilitate reproducibility.

- Overall, the study introduces an interesting approach, but the claim that metaheuristics outperform deep learning requires further validation. A key concern is the high recall but low precision tradeoff, which is not sufficiently explained.

Experimental design

see, above

Validity of the findings

see, above

Additional comments

Authors may want to have a look into: https://arxiv.org/abs/2407.02122
as it is about the datasets and may be useful


Cite this review as

Reviewer 5 ·

Basic reporting

In this work the authors describe a multi-objective optimization method for the detection of fake news that simultaneouly optimizes precision and recall. The proposed approach is based on the combination of text analysis methods and the NSGA-2 algorithm.

Experimental design

The proposed method was tested on three datasets: Covid-19, Syrian war and general news. The proposed methods found fast solutions in low dimensional data sets compared with other machine learning methods, and it stood out especially with its ability to optimize recall values.

Validity of the findings

The work is interesting but the paper needs to be improved in some of its parts. For instance, the three datsets should be explained more in detail and some statistics should be given. Moreover, it is not clear whether or not the general news dataset is an in-house dataset. It would have been interesting if the authors could have used a well-known dataset such as the one of Rashkin et al. (2017) that has been used in many other works. This could make possible a direct comparison with the results previously obtained. I would suggest the authors to carry out some experiments on this dataset as well.

Rashkin, H., E. Choi, J. Y. Jang, S. Volkova, and Y. Choi. 2017. Truth of varying shades: Analyzing language in fake news and political fact-checking. In Proceedings of the 2017 conference on empirical methods in natural language processing, pages 2931–2937

Additional comments

As future work, it would be interesting if the authors could apply their method on the specific case of conspiracy theories. Maybe they could use the datset of texts of Telegram that was introduced by Korencic et al. (2024).

Last but not least, the section on the related works should be improved and it should be enriched with recent publications where Transformers and Large Language Models have been employed to detect disinformation in texts. Moreover, recents surveys on fake news detection should be also mentioned, e.g. (Ruffo et al., 2023)

The paper is quite well written but it could be improved in some of its parts:

line 48: unecessary space at the beginning of the line
147/148: something should be said about the proposed method for automatically detect fake news (all the proposed methos aim to do it and it should be said how this is done in this case)
181: and NSGA-2 algorithm -> and the NSGA-2 algorithm

Korencic D., Chulvi B., Bonet X., Taul M., Toselli A.H., Rosso P. What Distinguishes Conspiracy from Critical Narratives? A Computational Analysis of Oppositional Discourse. In: Expert Systems https://doi.org/10.1111/exsy.13671

Ruffo, G., A. Semeraro, A. Giachanou, and P. Rosso. 2023. Studying fake news spreading, polarisation dynamics, and manipulation by bots: A tale of networks and language. Computer science review, 47:100531

Cite this review as

---

## Round 0.2 · Major Revisions

Two concerns remain despite your revision. The datasets used in your work are small and do not represent real-world scale or complexity. Claims of novelty and superiority are without statistical or methodological rigor (which is, in part, linked to the dataset concern).

For the work to be publishable, more extensive experiments with larger and wider variety of data corpus, to demonstrate and benchmark more appropriately the efficacy of your approach is necessary.

**Language Note:** The review process has identified that the English language must be improved. PeerJ can provide language editing services - please contact us at [email protected] for pricing (be sure to provide your manuscript number and title). Alternatively, you should make your own arrangements to improve the language quality and provide details in your response letter. – PeerJ Staff

Reviewer 3 ·

Basic reporting

The manuscript presents a general structure that aligns with academic standards, including an abstract, introduction, methodology, experiments, and discussion. However, several areas require significant improvement to meet the journal's expectations:

Language and Grammar: The manuscript contains numerous grammatical and stylistic errors. Phrasing is often verbose or repetitive, which detracts from readability. A full professional proofreading is highly recommended.

Figures and Tables: Key figures and visualizations were missing in the original manuscript. While these appear to have been added later, proper in-text references and captions are needed. Tables should include standard deviation values where appropriate (e.g., for cross-validation results).

Literature Review: While the Related Works section was added after revision, it still lacks depth and clarity in critically analyzing recent state-of-the-art approaches (especially transformer-based models). Many claims are not backed by strong comparative discussion.

Terminology Usage: Some terms are used inconsistently or inaccurately. For instance, the paper refers to TF-based weighting as "double normalized K-term frequency" without clear alignment with conventional definitions. Please ensure terminology is correct and consistently applied.

Experimental design

The experimental section demonstrates an intent to validate the proposed approach, but falls short in several critical areas:

Dataset Limitations: All experiments are conducted on small or niche datasets. There is no testing on well-known benchmark datasets (e.g., LIAR, FakeNewsNet), limiting the study’s relevance and generalizability.

Baselines: The chosen baselines (SVM, NB, DT, etc.) are basic and do not reflect current state-of-the-art in fake news detection. The lack of comparison with recent transformer-based models (e.g., BERT, RoBERTa) significantly weakens the empirical evaluation.

Parameter Justification: There is minimal justification for the choice of hyperparameters or why standard values were used. No hyperparameter tuning process is documented.

Ablation Studies and Error Analysis: The paper lacks any ablation study to isolate the contribution of the proposed CDL method. No sensitivity analysis or robustness testing is performed.

Interpretability: Although the paper claims interpretability via Pareto solutions, no example of actual model decisions or human-interpretable results is provided.

Validity of the findings

The core findings of the study are not sufficiently validated due to the following concerns:

Limited Dataset Size: Given the small size of the datasets used, the claims of superior performance must be interpreted cautiously. Results may not generalize to larger or more diverse real-world datasets.

Overfitting Risk: Recall values of 1.0 reported in some results suggest possible overfitting. While cross-validation was eventually performed, no statistical significance testing (e.g., t-tests) was provided to confirm robustness.

Comparative Weakness: Without benchmarking against modern deep learning models or more diverse datasets, the effectiveness of the proposed method cannot be confidently asserted.

Insufficient Novelty Validation: While the CDL modification is positioned as innovative, it lacks thorough empirical validation to demonstrate its advantage over traditional CD in various settings.

Additional comments

The overall contribution of the manuscript is limited by the narrow experimental scope and lack of benchmarking against more sophisticated models.

While the idea of optimizing both precision and recall via multi-objective optimization is conceptually sound, it is not new in the context of machine learning, and its "first time" application in fake news detection does not provide enough innovation in itself.

The manuscript would benefit from a clearer positioning within the existing literature and a stronger justification of the chosen methodological trade-offs.

I strongly recommend the authors revisit the experimental framework, strengthen the comparison with state-of-the-art models, and enhance the clarity and rigor of the writing.

Cite this review as

Reviewer 4 ·

Basic reporting

.

Experimental design

.

Validity of the findings

.

Additional comments

*
Verified github link is no longer broken and that would be nice if
authors added information where to access the dataset for further
reproduceability
*
It would be beneficial to include a dedicated subsection for the
dataset. Please consider referring to the organization style used in the
following papers for guidance:
https://ieeexplore.ieee.org/document/10704605 [1]
https://arxiv.org/abs/2407.02122.

Cite this review as

---

## Round 0.3 · accepted · Accept

The reviewers are satisfied with the revision.

Reviewer 4 ·

Basic reporting

This is 3rd round review. All is OK now -- from our perspective.

Experimental design

This is 3rd round review. All is OK now -- from our perspective.

Validity of the findings

This is 3rd round review. All is OK now -- from our perspective.

Additional comments

This is 3rd round review. All is OK now -- from our perspective.

Cite this review as